# Computer Vision Applied to Detect Lethargy through Animal Motion Monitoring: A Trial on African Swine Fever in Wild Boar

**DOI:** 10.3390/ani10122241

**Published:** 2020-11-29

**Authors:** Eduardo Fernández-Carrión, Jose Ángel Barasona, Ángel Sánchez, Cristina Jurado, Estefanía Cadenas-Fernández, José Manuel Sánchez-Vizcaíno

**Affiliations:** 1VISAVET Center and Animal Health Department, Veterinary School, Universidad Complutense de Madrid, 28040 Madrid, Spain; cjdiaz@ucm.es (C.J.); estefaca@ucm.es (E.C.-F.); jmvizcaino@ucm.es (J.M.S.-V.); 2Computing and Artificial Intelligence, Computer Science School, Universidad Rey Juan Carlos, 28933 Móstoles, Spain; angel.sanchez@urjc.es

**Keywords:** infectious disease, computer vision, artificial intelligence, african swine fever

## Abstract

**Simple Summary:**

African swine fever threatens pig welfare worldwide. Among its clinical signs, this disease manifests fever and weakness followed by progressive deceleration in the animal activities. The current computer vision advances allow us to detect animals, to track their movements and, therefore, to monitor animal activity. In this work, we used this technology to compute animal motion in a trial with animals infected with African swine fever virus, and proved a significant reduction in motion when the body temperature increased.

**Abstract:**

Early detection of infectious diseases is the most cost-effective strategy in disease surveillance for reducing the risk of outbreaks. Latest deep learning and computer vision improvements are powerful tools that potentially open up a new field of research in epidemiology and disease control. These techniques were used here to develop an algorithm aimed to track and compute animal motion in real time. This algorithm was used in experimental trials in order to assess African swine fever (ASF) infection course in Eurasian wild boar. Overall, the outcomes showed negative correlation between motion reduction and fever caused by ASF infection. In addition, infected animals computed significant lower movements compared to uninfected animals. The obtained results suggest that a motion monitoring system based on artificial vision may be used in indoors to trigger suspicions of fever. It would help farmers and animal health services to detect early clinical signs compatible with infectious diseases. This technology shows a promising non-intrusive, economic and real time solution in the livestock industry with especial interest in ASF, considering the current concern in the world pig industry.

## 1. Introduction

The applications of computer vision currently reach a wide range of disciplines in research and industry. Among the most popular, we find face recognition, virtual reality, video surveillance, handwriting recognition, image object search, pedestrian identification and self-driving cars. In the livestock industry, great efforts have been made to improve production characteristics, such as pig weight and body size measurement [1,2], animal counting [3], thermal comfort [4,5], meat quality classification [6,7], etc. In recent years, the popularity of computer vision in animal recognition and tracking has increased considerably, specially with Artificial Intelligence algorithms [8,9,10]. These developments are of particular interest in veterinary epidemiology when focusing on measuring behavioural changes in animals for health and welfare [11,12].

African swine fever (ASF) is an infectious disease of swine that affects domestic pigs and wild boars in Europe. Different clinical presentations have been described depending on the virulence of the isolate, host species, breed and route of infection [13]. Severe clinical forms are characterized by high fever and mortality rates up to 100% within 4–9 days. Mild forms produce lighter symptoms and mortality rates ranging between 30 and 70% [14]. Other isolates can induce subclinical or even unapparent forms. Its notification is mandatory to the World Organisation for Animal Health (OIE) mainly due to mortality rates and great sanitary and socioeconomic impacts in affected territories.

Since 2007, ASF has spread through Eastern European countries even reaching Central and Western Europe. The first outbreak was reported in Georgia from where it rapidly spread to the Russian Federation (2007). Later, Ukraine (2012), Belarus (2013), Estonia, Lithuania, Latvia and Poland (all 2014) reported infection, mainly in wild boar populations. Then, six European countries became infected: Moldova (2016), Romania (2017), Czech Republic (2018), Hungary (2018), Bulgaria (2018), Belgium (2018) and Germany (2020), see Figure 1. It was reported in Asia for the first time in August 2018. Elsewhere, in China, which contains almost half of the world domestic pig population, the disease has spread very quickly with more than 100 outbreaks from 28 different provinces and around one million of domestic pigs have been culled [15]. Currently, Mongolia (2019) and Vietnam (2019) have reported ASF (OIE, www.oie.int). This situation highlights the great importance of early detection of ASF in order to control its spread.

The challenge for minimizing ASF spread focuses on the early detection especially in the first stages of the clinical symptoms of the animals. Here, propose the use of artificial vision technologies to monitor the animal behaviours. As expected, this technology can be implemented in indoors or enclosures but probably not feasible in the free ranging wild boar populations. However, we hope this technology will help for the rapid detection of ASF in indoor farms, which can be helpful to control the spread of the disease and therefore minimize the economic costs which threatens the swine industry [16,17].

Generally, infectious diseases manifest fever and weakness followed by progressive deceleration in diurnal activities. This change in movement patterns might be difficult to detect by simple observation in early stages of infection. However, previous works demonstrated that it can be detected by quantifying animal movements through new technologies [18,19]. Computer vision algorithms were able to detect the progressive reduction in the overall motion of animals infected with ASF virus before severe clinical signs [19]. Because ASF is a rapidly spreading disease, the critical challenge is reducing the time of disease detection. In this work, we propose video monitoring to compute animal motion patterns associated to higher body temperatures. Thus, an improved algorithm could be able to trigger alerts of suspicion once the first infected animal shows a significant reduction in movements patterns.

In this work, we aim to improve the previous algorithm by including Artificial intelligence and Kalman filter techniques with the goal of tracking precise motion of animals individually. The goal in this work is to show that animal motion behaviours change under high body temperatures caused by ASF virus. Thus, two experiments were carried out under video surveillance where the here developed algorithm monitored the motion.

## 2. Materials and Methods

### 2.1. Ethics Statement

The experiments were carried out at the laboratory BioSafety Level 3 (BSL-3) of the VISAVET Centre at the Veterinary Faculty of the Universidad Complutense de Madrid (UCM; Madrid, Spain), between May and June 2018. The experimental protocol was approved by the Ethic Committees of Animal Experimentation following European, National and Regional Law and regulations by the reference number PROEX 124/18 (Regulation 2010/63/EU and the Spanish Royal Decree 53/2013). The protocol included a detailed description of efforts to prevent unnecessary suffering of the animals, including humane endpoints, anesthetic procedures [20] and euthanasic protocols following Garrindo-Cardiel et al. clinical evaluation guidelines [21]. Specifically, human endpoints were reached when at least one parameter was scored as severe. Animal’s health and welfare was checked daily by a clinical veterinarian. Animals were monitored 24 h/day by videocameras which were regularly checked.

### 2.2. The Experiments

Two experiments were carried out by monitoring a pen surface with fixed dome cameras ceiling-mounted. The cameras were configured to record 15 min of standalone video sequences of 640 × 360 pixels in RGB format in 6 frames per second, approximately; i.e., around 5400 frames per each 15-min-video sequence. The cameras recorded in diurnal and nocturnal mode. However, only diurnal mode was processed in this work that approximately represented 63% of the total recorded time. Animals used in these two experiments were Eurasian wild boar of 4 months old. In order to track animal movements, wild boar wore blue and orange colored neck collars during the whole experiments.

The first experiment (E1) lasted 16 days in a pen surface of around 2 m${}^{2}$. Here, two wild boar were isolated in the pen and, subsequently, infected with a virulent, haemadsorbing, genotype II ASF virus (ASFV) isolate (Arm07). Afterwards, the animal activity were recorded with only one camera. One animal wore an orange collar around the neck and, the other one, a blue collar. Since the first day of the experiment (E1D1, Day 1 of Experiment 1), the animals were continuously monitored with video cameras until the last day of the experiment (E1D16). A total of 242 h were recorded in diurnal mode. The algorithm was developed and fitted for tracking both animals individually and computing motion on the fly. The main goal of this experiment was to find out whether low motions are correlated with high temperatures caused by fever.

The second experiment (E2) lasted 22 days in a pen surface of around 5 m${}^{2}$. Here, two wild boar were infected with a virulent ASFV isolate, Arm07. Both animals wore an orange collar. In addition to them, four wild boar were previously inoculated with an attenuated, non-haemadsorbing, genotype II ASFV isolate, named Lv17/WB/Rie1 vaccine candidate [22]. This isolate was used as a vaccine in order to prevent infection and clinical course caused by contacting wild boar infected with Arm07. These four animals were collared in blue. Since the first day of the experiment (E2D1, Day 1 of Experiment 2), six animals were continuously monitored with video cameras until the last day of the experiment (E2D22). A total of 332 h were recorded in diurnal mode. The algorithm was fitted for tracking multiple animals and computing the total motion per group on the fly. This experiment was designed to test different motion patterns between vaccinated (healthy) and infected animals.

### 2.3. Algorithm

The first target of the algorithm is to recognize each blue or orange colored collar on a single frame (*animal recognition*). The second objective is to track each collar along consecutive frames (*animal tracking*). The final objective is to compute collar displacements along all frames of any video sequence. All code was developed in Matlab 2018.

#### 2.3.1. Animal Recognition

The cameras were recording a similar background in both experiments: a green gridded floor fenced in gray bars, black eating troughs and gray drinking troughs (see Figure 2, left). As animal’s fur was brown, so orange and blue collars contrasted with the background which eased their recognition. Here, the objective of the algorithm is to recognize blue and orange collar in a single frame.

For this purpose, three consecutive processes are carried out. The first process consists on a binary image segmentation based on color threshold. In other words, pixels with similar RGB values to blue/orange are filtered in a single frame (see Figure 2, right). Thus, pixels from targeted collars are filtered. Then, most pixels associated to other color-based objects were discarded, except some residual pixels with similar color to blue/orange.

The second process involves morphological operations and blob analysis in order to remove residual pixels. Moreover, pixels from objects which have been occluded due to light and shade effects are connected (see Figure 3); they are called *connected components*. Such pixels are generally associated to unique objects or color-based parts of a specific object, hereinafter they will be referred to as objects. This process allows discarding of irrelevant objects in the scene, minimizing the computational cost. Furthermore, once every filtered object in the frame is located, the blob analysis retains additional valuable information such as coordinates of the object (i.e., its centroid) which are needed for motion estimation.

The third process focuses on object classification; i.e., to differentiate a collar from any other objects in the scene. Here, deep learning Convolutional Neural Networks (CNNs) have revolutionized the state of object recognition in recent times [23]. CNNs allow the discovery of complex structures in large data sets, such as images. A detailed description of the CNN working can be found easily in the literature [24,25]. Briefly, CNNs are mainly composed by a stacked layer architecture and parameters trained to connect them.

Although the training of CNNs is computationally expensive and the design of the CNN architecture is complicated, several pre-trained CNNs for object recognition are publicly available. The transfer learning is the process of training only some key layers from these CNNs in order to make it useful to detect objects of interest. In this work, we used a very popular CNN for object recognition, called AlexNet [26], that was re-trained for the recognition of the blue and orange collars.

AlexNet was trained and tested with input images of 227 × 227 × 3 resolution. These images were randomly chosen from the first two days of video sequences from E1. Concretely, every detected object in the scene was boxed into a picture of 113 pixels, up and down along vertical and horizontal coordinates, from its centroid (as the yellow boxes plotted ones in Figure 2 and Figure 3). A final dataset of 1000 images was afterwards labeled: 300 pictures of blue collar, 300 pictures of orange collar, 250 images of other objects and 150 images associated to sanitizing and sampling activities carried out by veterinarians (see Figure 4). This dataset of images was randomly divided into a training set (80%) and a test set (20%) for AlexNet training.

#### 2.3.2. Animal Tracking

Animal tracking consists of locating the position of the animal along consecutive frames. In both experiments, we tracked the centroid of each collar. In E1, animal tracking could be computed easily because we conducted this experiment with two different collars. In E2, animal tracking became more difficult because six animals would imply six different color-based collars. In addition, these collars should be different to each other, and different enough with respect to the background and animal’s fur. Finally, we split the herd into two groups, infected and healthy ones. But the new challenge was to track more than one same coloured collar within the scene.

The Kalman Filter (KF) is a widely used technique for tracking objects. Essentially, KF aims to predict the future location of objects; i.e., KF will predict the animal position on the next frame {t+1} based the previous locations {⋯,t−2,t−1,t}; henceforth, KF-Prediction. Nevertheless, KF needs to assume initial parameters associated with Gaussian noise or error thresholding. The KF-Prediction was configured in Matlab through *configureKalmanFilter* command.

Although KF could predict future positions, the algorithm might not find any connected components in that position when computing the next frame (due to occlusion or light changes). In this case, KF configuration uses the solution proposed by the Hungarian assignation problem resolution [27]; henceforth, KF-Assignation. In few words, the algorithm assigns the position of the collar to the closest connected component, only if this component was not assigned to any other residual objects or if this component was not excessively far from the predicted location.

#### 2.3.3. Motion Computation

For two consecutive frames, at times t−1 and *t*, the centroid of the collar located on $\left(x_{t-1},y_{t-1}\right)$ and $\left(x_{t},y_{t}\right)$, respectively, the motion is computed by the Euclidean distance:(1)$$\begin{array}{c} m_{t,t-1}=\sqrt{{\left(x_{t}-x_{t-1}\right)}^{2}+{\left(y_{t}-y_{t-1}\right)}^{2}}. \end{array}$$

Thus, the total motion of one collar throughout a whole video sequence, *v*, is computed as the sum of all motions $m_{t}$ along all frames of the video N∈N; i.e.,:(2)$$\begin{array}{c} m_{v}=\sum_{t=2}^{N}m_{t,t-1}. \end{array}$$

Consequently, the total motion carried out for each animal in a whole 15-min-video sequence *v* is computed as the sum of all movements between frames as:(3)$$\begin{array}{c} m_{d}=\sum_{v\in D}m_{v}, \end{array}$$
where D is the set of videos regarding a day d∈{D1,⋯,D16} for E1 or d∈{D1,⋯,D22} for E2.

#### 2.3.4. Algorithm Overview

Figure 5 shows a workflow overview of processing a video sequence, *v*, frame by frame, being N∈N the total of frames. The algorithm processed step-wise every frame and stored key information into a log file.

For each frame t∈{1,⋯,N}, the first task of the algorithm is to filter orange/blue pixels in the scene (*Color Threshold*), as shown in Figure 2. Next, the algorithm computes morphological operations and blob analysis in order to create color-based objects, which are the candidates representing targeted collars (*Connected components*), as shown in Figure 3.

In the next steps, the algorithm needs to find the spatial location of the collar on the frame *t*, $\left(x_{t},y_{t}\right)\in R^{2}$. To do this, the algorithm associates the targeted collar to one of the connected components previously identified. Subsequently, it uses the CNN to classify the connected components as collars or not collars, and extract the centroid position of the collar. However, as aforementioned, it implies two important challenges, first, the multiple tracking assignation, and second, the computational costs, but KF solves these difficulties. Thus, only if the algorithm is processing the first frame, t=1, or KF-Assignation couldn’t assign any connected component to the collar in the previous step, CNN tries to recognize the collar among all connected components previously filtered (*Collar Recognition*). Otherwise, KF-Assignation assigns one of the connected components to the collar.

Once located the collar in the scene, the algorithm computes the position of the centroid of the collar (*Location*
$\left(x_{t},y_{t}\right)$) and computes the displacement between the previous frame location and the current one for t>1 (*Motion $m_{t,t-1}$*), as in Equation (Equation 1).

In the final step, previous to process the next frame (t:=t+1), the algorithm computes the future position of the collar through KF-Prediction based on the previous spatial positions and on the configuration of KF (*Predict Location*
$\left(x_{t+1},y_{t+1}\right)$).

### 2.4. Statistical Analysis

The goal of E1 was to test the correlation between high temperatures and motion reduction. Along this experiment, some specific days were selected for temperature sampling according to sanitary protocols [21]. The coefficient correlation between temperature and motion was computed for repeated measures from two animals. Here, we used the R package *rmcc* [28].

E2 was focused on testing significant differences in motion computation between healthy (vaccinated animals) and infected animals (animals infected with Arm07). A non-parametric Wilcoxon Signed Rank test was carried to test daily differences in motion scoring between both groups.

In both experiments, daily motion was computed to each colored collar. E1 was focused on single tracking, therefore individual tracking was carried out and total daily motion was estimated for each animal. In E2, several animals wore collars of the same color, and individual tracking could not be computed. In this experiment, daily motion was averaged by the number of animals wearing collars of the same color: ${\hat{m}}_{d}/2$, orange collars for infected, and ${\hat{m}}_{d}/4$, blue collars for healthy.

## 3. Results

### 3.1. Algorithm Evaluation

The CNN was trained with the Stochastic Gradient Descent (SGD) algorithm configured with a learning rate of 0.001, a maximum of 20 epochs and a batch size of 64. The final accuracy classifying blue and orange collars with images from the test set scored 97.2% and 95.40%, respectively.

The KF-Prediction was fitted under a constant velocity pattern with an initial estimated error variance, in location and velocity, of 200 and 50 pixels, respectively. In addition, the tolerance of KF for deviation constant velocity model, in location and velocity, was 100 and 25 pixels, respectively. With this configuration, after fitting parameters on the first frames, KF was able to predict future positions any target object in motion in combination with KF-Assignation based on a cost matrix built-in from Euclidean distance between connected components. A sample result obtained after tracking the blue collar under this configuration can be displayed in Appendix A. Furthermore, a sample result after tracking the orange collar and its motion computation can be displayed in Appendix A.

### 3.2. Statistical Analysis

Figure 6 shows the total daily motion $m_{d}$ computed in both experiments (units of motion: pixels per frame). Blue/orange bars represented the total motion computed every day per each animal in E1, and the averaged motion of the animals within each group in E2.

Results of E1, where animals were individually monitored, showed that both animals had a similar behavior in motion values along the experiment (except day D8 and D10). Low values of daily motion occurred twice, between D4–D6 and D11–D14. After D15, both animals showed an increment in motion activity.

As E1 aimed to comparatively test motion and rectal temperatures of each animal, rectal temperature was monitored on D1, D4, D6, D11, D13, D14, D15 and D16 days. Table 1 shows the temperatures values on sampling days and daily motion of each animal. High variation of temperature values (between 38–41 °C) were recorded across the study period. Values higher than 40.5 °C were likely owing to viremia peaks during ASFV infection, which were confirmed by PCR analysis. As expected, low values in daily motion matched with high temperature peaks.

According to results showed in Table 1, the correlation coefficient between temperature and motion scores −0.58, with 13 degrees of freedom (from eight samples per animal).

Results E2 showed higher and more established daily motion values. This experiment aimed at comparing motion behavior between ASFV infected animals (orange collar) and healthy animals (blue collar). As shown in Figure 6, blue values were generally higher than the orange ones. However, results of Wilcoxon Signed Rank test did not reject the hypothesis that animals infected with virulent ASFV isolate had significant differences in motion with regard to healthy animals (*p*-value <0.01).

When comparing both experiments, results from E1 and E2 strongly differed in total daily motion values; i.e., while E1 motion values ranged [5, 30] × 1000, E2 values varied around [25, 100] × 1000. Nevertheless, statistical tests could not be applied to test differences, as their goals and biological trials differed.

## 4. Discussion

### 4.1. On the Methodological Approach: Computer Vision Application in Animal Motion Monitoring

Remote observation technologies based on computer vision have the potential to be a reliable and efficient support for monitoring scenes and execute algorithms beneath in real-time for different approaches. These technologies are able to automatically control the animal behavior surveillance in controlled systems, such as farms, where multiple parameters can be determined by individual motion. In agreement with previous studies based on motion-based video analyses [19], our results suggest that computer vision systems can be useful to detect changes in animal movement patterns. As previously referred, these changes in animal movement may indicate presence of clinical symptoms such as fever.

Although the outcomes showed a negative correlation between movement temperature, this work has potential limitations. First, sampling size should be higher, however, the complex biosecurity measures involving ASFV difficult massive studies. Second, these experiments have been carried out under experimental conditions, thus real scenarios, as indoor farms, may differ, but animal motion monitoring has been already tested in similar studies with successful results [8,9,10]. Third, a significant decrease in daily motion may not be a fever indicator, they might be other factors such as climatic conditions or human interaction, though a significant motion behaviour is worth monitoring even for these factors. And fourth, the animals have been monitored through necklaces that should be removed in real scenarios. Future work should fit the algorithm for animals not wearing collars. It can be conducted by applying the new state-of-the-art algorithms for object detection and tracking. In this work, we aimed to compute the animal motion in experimental conditions and demonstrating a deceleration on motion when they were infected. Thus, we reduced the complexity of developing animal identification algorithms and we focused only on motion computing. However, on the basis of this algorithm, future work would focus on individual identification based on animal shapes [29] that ease the nocturnal monitoring and it also could track animals with infrared (IR) or thermal spectra cameras.

In this work we used the neural network of AlexNet and Kalman filter as the techniques that better fitted to our identification and tracking approaches. However, other options could have worked with the studied scenario. For instance, ResNet50 and VGG16 were also explored as potential methods to classified objects of interest. The obtained results for such techniques were very similar to AlexNet. Therefore, we decided to use AlexNet considering that (i) results were homogenous among the three techniques and (ii) AlexNet was well documented and implemented in Matlab. Almost for the same reason, we chose Kalman filter as tracking strategy.

### 4.2. Motion Patterns and ASFV Infection Surveillance

Testing animal motion patterns may be a valuable alternative to traditional sentinel sampling. Our motion results correlated ASF clinical course, mainly represented by fever (rectal temperature) at individual (E1) and collective level (E2). These results align with literature regarding animal motion and ASFV infection on pigs [18,19]. Overall, these results can be useful to detect animals with high body temperatures weakening. For instance, it could be implemented to detect animals affected with any infectious pathogens with especial interest in transboundary diseases such as ASF, foot-and-mouth disease, classical swine fever, blue-tongue, Rift Valley fever or African horse sickness. In this sense, our results showed a significant relationship between decreases in animal motion and presence of disease before showing evident clinical signs and pathological findings.

The next objective of this study should be to tackle the real implementation of this system in indoor farms. The implementation of this technology in outdoors needs likely further work. But, even focusing on the detection of ASF rapidly only in indoor farms, the fight against the spread of ASF would advance considerably. The upwards digitization of current farms in Europe shows the ideal scenario for implementing affordable video monitoring systems as introduced here. In fact, approximately 74% of European pig production is carried out in indoor pig farms [30], where these monitoring tools would be possible. The farm’s infrastructure to implement this technology would need video-surveillance capabilities to monitor pig pens (preferable not massive) with ceiling-mounted cameras. Nowadays, every video surveillance system usually uploads movie clips to the cloud where they can be collected for motion computation through this system in real or nearly real-time.

The rapid spread of ASF -in three different continents, affecting more than 55 countries and more than 77% of the world swine population- threatens the pig industry worldwide. The quest of methods to early detect ASF, that causes rapid clinical courses in which the only initial indication would be fever in few animals, might help the current epidemiological and control procedures.

## 5. Conclusions

This research provides a innovative approach to explore new strategies to detect infectious diseases early, which is crucial to minimize economic and health consequences of uncontrolled spread. This study has taken advantages from innovative technologies previously implemented in other fields, deep learning and computer vision, to detect lethargy of ASF among wild boars under experimental conditions. We have shown that changes in motion patterns on infected individuals with ASF virus and infected groups differ from healthy ones through video monitoring. This proves that working towards computer vision algorithms in veterinary epidemiology promises to be potentially useful to real-time ASF surveillance.

## Figures and Tables

**Figure 1 animals-10-02241-f001:**
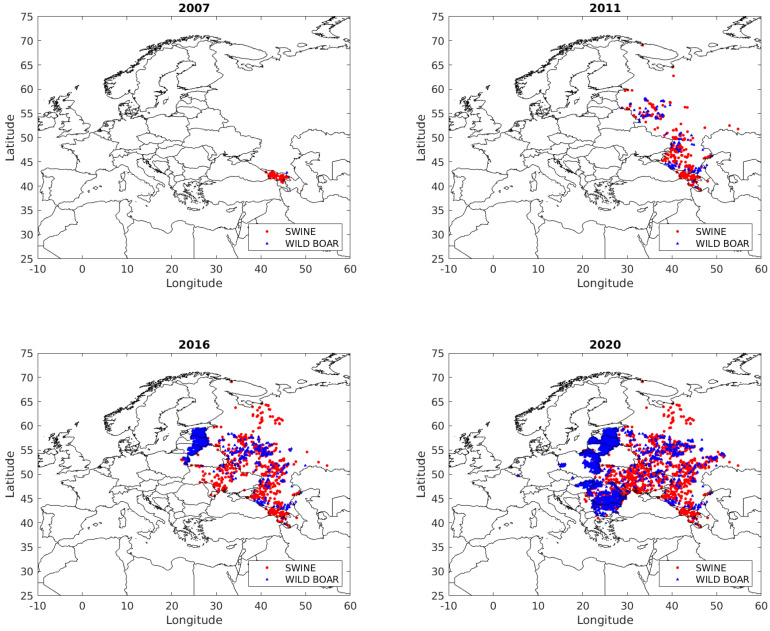
Evolution of ASF outbreaks in Europe since 2007 to 2020: Notified outbreaks of ASF in Europe (except Sardinia, Italy) summarized on years 2007, 2011, 2016 and 2020 (October).

**Figure 2 animals-10-02241-f002:**
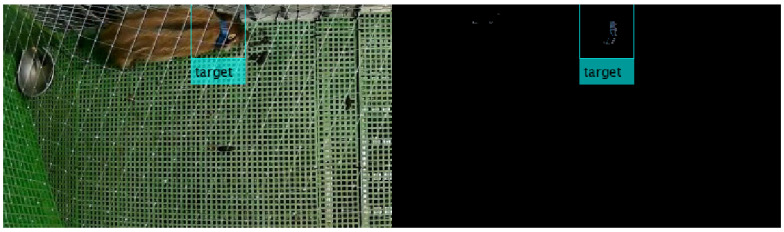
Random frame of fixed camera recording scene. Color threshold. (**Left**) Real frame corresponding to E1, where only two wild boar wore collars. (**Right**) Blue RGB filter applied to the same frame.

**Figure 3 animals-10-02241-f003:**
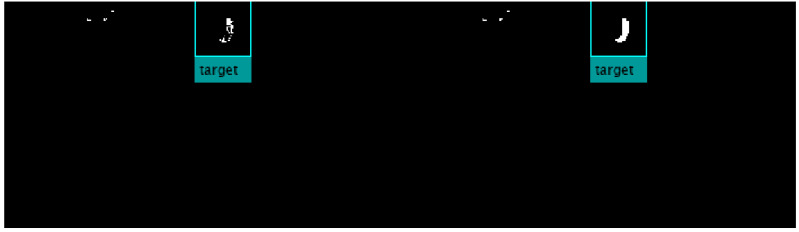
Before and after morphological operations and blob analysis on the binary mask. Left: Same random frame after blue threshold binary mask. Right: Resultant frame after morphological operations and blob analysis.

**Figure 4 animals-10-02241-f004:**
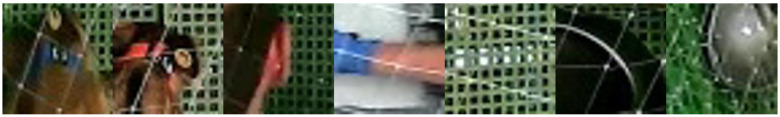
Samples of 227 × 227 × 3 pictures for training AlexNet. From left to right: orange collar, blue collar, ear of a veterinarian, a forearm, gridded ground and eating/drinking drinking troughs. All these objects show blue/orange connected pixels, then objects to be classified.

**Figure 5 animals-10-02241-f005:**
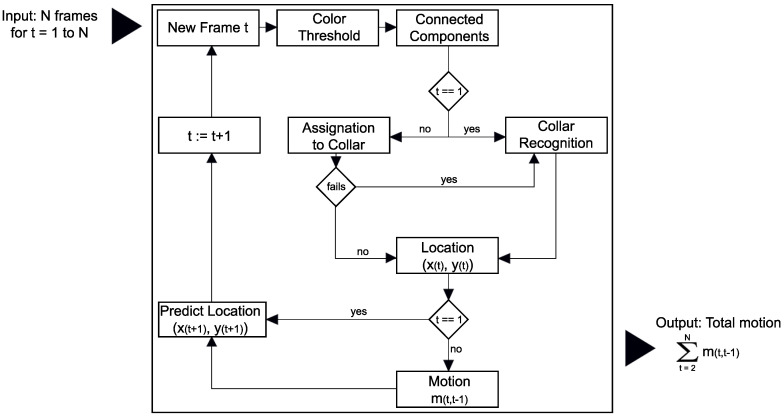
Main workflow of the algorithm. Step-wise processing of all video frames including Object Recognition, Object tracking and Motion computation.

**Figure 6 animals-10-02241-f006:**
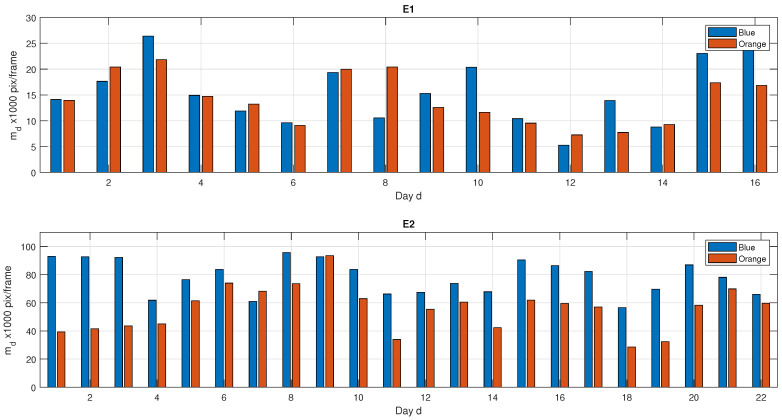
Daily motion along both experiments, E1 and E2. Daily motion $m_{d}$ computed in each experiment for animals with blue/orange collars. Top: both animals are infected with ASFV Arm07. Bottom: orange specimen (two animals) were infected with ASFV Arm07, blue specimen (four animals) were vaccinated with Lv17/wb/Rie1.

**Table 1 animals-10-02241-t001:** Temperature and motion scores in E1. Temperature (°C) and daily motion (×1000 pix/frame) on sampling days for each animal (both infected with ASFV Arm07).

Day	Blue Temp.	Orange Temp.	Blue Motion	Orange Motion
1	39.80	39.96	15.96	13.95
4	40.60	40.15	14.74	14.92
6	40.57	40.68	9.10	9.61
11	40.58	40.39	9.54	10.40
13	40.09	39.98	7.73	13.89
14	39.07	40.82	9.25	8.77
15	38.70	40.00	17.35	24.66
16	38.30	39.20	16.84	22.89

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
