# Peer review of "Computer Vision Applied to Detect Lethargy through Animal Motion Monitoring: A Trial on African Swine Fever in Wild Boar"

_animals, 2020, doi:10.3390/ani10122241_

Round 1
Reviewer 1 Report
I think this is an interesting and useful study. Congratulations to the authors. I have no comments on the content, which seems correct to me. Most of my comments are minor, basically suggestions on structure and grammar and some on figs. I am not a native English speaker, but still, I think the text requires some review.
A couple of questions/comments, mainly out of curiosity and from a practical perspective:
-would a thermal camera serve to detect fever? Or fever is only noticeable in pigs via rectal and not detectable on the body/skin/fur?
-In line with the last paragraph of the ms. I would like to read some general numbers to get an idea of the “potential market” or applicability range of this approach (how important can this study be). For example, just considering pigs, I d like to know how many (%?) are captive (intensive farming, indoors) so that indoor cameras and this approach are feasible and what s the % of extensive (so this may not be an option). Maybe not just for pigs but for farming, as you mention this could serve to detect other illnesses or conditions. E.g. are 80, 50, 5% of the world domestic animals indoors so they could be monitored by cameras and IA?
Title: I suggest using African swine fever instead of ASF. I think it s better to avoid the acronym on titles.
L8: Lower better than low?
L25 use “that” instead of “which”.
L25-6 Check plural concordance: Use either pigs and boars or pig and boar
L37 “Since August 2018, ASF was reported for the first time in Asia”, better use: “it was reported in Asia for the first time in August 2018” or “It is present in Asia since August 2018”
L38 use “half”; not “the half”
L50 “Despite the ASF is a rapidly spreading disease, next challenge is to reduce the time of disease detection by monitoring the motion of every animal individually.” Review the phrase. Maybe “ because ASF is a rapidly spreading disease, the current challenge is reducing the time of disease detection” and then mention that you suggest doing it “by monitoring the motion of every animal individually”, as I think there may be other alternatives
After reading the abstract and introduction I am still wondering how are you suggesting to obtain the images that you plan to use for computer vision/IA or the data for deep learning. I see it in methods but I think it should be mentioned in the intro.
Methods: text should be reorganized for clarity. It does not flow, some phrases are not logically connected. E.g. “The second experiment lasted 22 days. Pen surface was around 5 m2 and two cameras were necessary. The previous day, six wild boar were isolated in the pen. Two of them were infected with a virulent ASFV isolate, Arm07. Both animals wore orange collar.” I think this could be substantially improved.
L66 “Animals were continuously monitored 24h/day by videocameras which were regularly
checked during the day” better: “Animals were monitored 24h/day by videocameras which were regularly checked”. Unless you want to highlight the cameras were not checked at night time.
L69 add “Internet Protocol” for IP and maybe also a link, maybe readers are not familiar with IP /CCTV
L69-73 could be better explained (simplified), review text
L73 daily, better use “diurnal” https://wikidiff.com/daily/diurnal
As adjectives the difference between daily and diurnal is that daily is quotidian, that occurs every day, or at least every working day while diurnal is happening or occurring during daylight, or primarily active during that time.
As nouns the difference between daily and diurnal is that daily is a newspaper that is published every day while diurnal is a flower that opens only in the day.
As a adverb daily is quotidianly, every day.
L76 4 months of age. Better 4 months old
L74 how many wild boars?
L81 “were continuously monitored with video cameras” but before you said just daytime
L99 target, here better use objective. https://bizfluent.com/info-8405368-differences-between-target-objectives.html
106 So, “the target of the algorithm is to recognize blue and orange collar in a single frame”. And just after, I see the image of Fig 2 where it seems to me that there are 2 animals and I think I can naked-eye see a blue collar (Id in blue) on the one on top and an orange collar on the one below (not indicated, recognized by the algorithm? ) Or you were just looking for blue there? Besides, I also see a big orange piece of something on the bottom of the figure. Please check what you re trying to show as it s currently confusing.
Fig 3 is not very impressive maybe adjust the scale so we can better see that the algorithm did something remarkable
L124 “A detailed description of the CNN working can be found easily in the literature.” Well, if so, please provide a ref…
L131 Apologies for my ignorance, but how is it that an image resolution has 3 dimensions? “227x227x3 resolution”
L137, sanitary or sanitizing?
Fig 4 an forearm use “a forearm”
L144 wouldn t have been easier juts using 6 different colors?
L245 I d say technological or innovative more than “multidisciplinary”
L249 proved, better than proofed
L251, you proved it, thus: “these systems CAN be used..” (not might)
L254 efficient support for doing what?
L254 “These technologies are able to automatically study” the technologies don t study. You study, the technologies monitor, record, register, detect, survey..
L259 Rephrase, grammar is wrong: “The obtained results shown a negative correlation between movement and temperature, however it may not always be so”
L280 these results align with literature…
Author Response
Dear reviewer,
We thank you for your deep revision. We really think that all your considerations have been very beneficial to the improvement of the manuscript. According to your comments, we have modified the content taking into account other two reviewers’ considerations. Finally, we answer your inquiries below.
REVIEWER: would a thermal camera serve to detect fever? Or fever is only noticeable in pigs via rectal and not detectable on the body/skin/fur?
AUTHORS: The authors have asked the same questions at some point of the study. The use of thermal cameras is, of course, one of the first options that we considered for this study. However, there were two issues that helped us to discard this option. First, the skin of animals (especially pigs) is quite thick, the ability to capture animal’s body temperature is quite poor. Second, the price of thermal cameras are excessively expensive compared to simple computer cameras. This project was carried out with approximately €50/camera. Compared with approximately €500/thermal-camera, it’d increase considerably the project. One of the goals of this work is the implementation of an economic system affordable for any farm.
REVIEWER: In line with the last paragraph of the ms. I would like to read some general numbers to get an idea of the “potential market” or applicability range of this approach (how important can this study be). For example, just considering pigs, I'd like to know how many (%?) are captive (intensive farming, indoors) so that indoor cameras and this approach are feasible and what is the % of extensive (so this may not be an option). Maybe not just for pigs but for farming, as you mention this could serve to detect other illnesses or conditions. E.g. are 80, 50, 5% of the world domestic animals indoors so they could be monitored by cameras and IA?
AUTHORS: The manuscript has been modified accordingly (Introduction and Discussion).
REVIEWER: Title: I suggest using African swine fever instead of ASF. I think it's better to avoid the acronym on titles.
AUTHORS: Done.
REVIEWER: L8: Lower better than low?
AUTHORS: Done.
REVIEWER: L25, use “that” instead of “which”.
AUTHORS: Done.
REVIEWER: L25-6 Check plural concordance: Use either pigs and boars or pig and boar
AUTHORS: Done.
REVIEWER: L37 “Since August 2018, ASF was reported for the first time in Asia”, better use: “it was reported in Asia for the first time in August 2018” or “It is present in Asia since August 2018”
AUTHORS: Done.
REVIEWER: L38 use “half”; not “the half”
AUTHORS: Done.
REVIEWER: L50 “Despite the ASF is a rapidly spreading disease, next challenge is to reduce the time of disease detection by monitoring the motion of every animal individually.” Review the phrase. Maybe “ because ASF is a rapidly spreading disease, the current challenge is reducing the time of disease detection” and then mention that you suggest doing it “by monitoring the motion of every animal individually”, as I think there may be other alternatives
AUTHORS: Thank you. Much better now.
REVIEWER: After reading the abstract and introduction I am still wondering how are you suggesting to obtain the images that you plan to use for computer vision/IA or the data for deep learning. I see it in methods but I think it should be mentioned in the intro.
AUTHORS: The intro explains now the implementation in indoor farms.
REVIEWER: Methods: text should be reorganized for clarity. It does not flow, some phrases are not logically connected. E.g. “The second experiment lasted 22 days. Pen surface was around 5 m2 and two cameras were necessary. The previous day, six wild boar were isolated in the pen. Two of them were infected with a virulent ASFV isolate, Arm07. Both animals wore orange collar.” I think this could be substantially improved.
AUTHORS: Modified.
REVIEWER: L66 “Animals were continuously monitored 24h/day by videocameras which were regularly checked during the day” better: “Animals were monitored 24h/day by videocameras which were regularly checked”. Unless you want to highlight the cameras were not checked at night time.
AUTHORS: Done.
REVIEWER: L69 add “Internet Protocol” for IP and maybe also a link, maybe readers are not familiar with IP /CCTV
AUTHORS: Due to it not being relevant for this study, we have removed “IP” from text.
REVIEWER: L69-73 could be better explained (simplified), review text
AUTHORS: Modified.
REVIEWER: L73 daily, better use “diurnal” https://wikidiff.com/daily/diurnal. As adjectives the difference between daily and diurnal is that daily is quotidian, that occurs every day, or at least every working day while diurnal is happening or occurring during daylight, or primarily active during that time. As nouns the difference between daily and diurnal is that daily is a newspaper that is published every day while diurnal is a flower that opens only in the day. As an adverb daily is quotidianly, every day.
AUTHORS: Thanks for this point. We have modified the text accordingly. Only, when talking about motion, we use ‘daily motion’, otherwise we use diurnal.
REVIEWER: L76 4 months of age. Better 4 months old
AUTHORS: Done.
REVIEWER: L74 how many wild boars?
AUTHORS: In both experiments, all animals were 4 months old. But the number of wild boars in every experiment was not the same. For this reason, we specify the number of wild boars in the next paragraph, where we explain the details of the experiments.
REVIEWER: L81 “were continuously monitored with video cameras” but before you said just daytime
AUTHORS: Yes, we monitor diurnal and nocturnal (continuously) but only diurnal is processed. The main reason is that in nocturnal mode, we can’t distinguish collars.
REVIEWER: L99 target, here better use objective. https://bizfluent.com/info-8405368-differences-between-target-objectives.html
AUTHORS: Done.
REVIEWER: 106 So, “the target of the algorithm is to recognize blue and orange collar in a single frame”. And just after, I see the image of Fig 2 where it seems to me that there are 2 animals and I think I can naked-eye see a blue collar (Id in blue) on the one on top and an orange collar on the one below (not indicated, recognized by the algorithm? ) Or you were just looking for blue there? Besides, I also see a big orange piece of something on the bottom of the figure. Please check what you are trying to show as it's currently confusing.
AUTHORS: In this case, we only evaluated the blue filter for detecting the blue collar. It has been clarified in text. Furthermore, we have improved the picture in order to zoom the animal.
REVIEWER: Fig 3 is not very impressive maybe adjust the scale so we can better see that the algorithm did something remarkable
AUTHORS: We have improved the picture in order to zoom the animal.
REVIEWER: L124 “A detailed description of the CNN working can be found easily in the literature.” Well, if so, please provide a ref…
AUTHORS: We have inserted the two famous papers explaining CNNs deeply.
REVIEWER: L131 Apologies for my ignorance, but how is it that an image resolution has 3 dimensions? “227x227x3 resolution”
AUTHORS: height x width x R-G-B (the three color channels). It is like 3 pictures of 227x227 that overlapped, resulting in the real coloured picture.
REVIEWER: L137, sanitary or sanitizing?
AUTHORS: Modified.
REVIEWER: Fig 4 an forearm use “a forearm”
AUTHORS: Done.
REVIEWER: L144 wouldn't have been easier just using 6 different colors?
AUTHORS: Totally agree. However, in pre-experiment tests, other colors do not have sufficient contrast with a green and gray background and the brown color of the animals' skin. In order to achieve this, there were few options left. And of the ones that were left, due to light and shadow, colors like red could look orange and get confused at some point. This would make the study lose sensitivity and, therefore, not achieve the individuality of the movement.
REVIEWER: L245 I d say technological or innovative more than “multidisciplinary”
AUTHORS: Done.
REVIEWER: L249 proved, better than proofed
AUTHORS: Done.
REVIEWER: L251, you proved it, thus: “these systems CAN be used..” (not might)
AUTHORS: Done.
REVIEWER: L254 efficient support for doing what?
AUTHORS: Remote observation technologies based on computer vision have the potential to be a reliable and efficient support for monitoring scenes and execute algorithms beneath in real-time for different approaches.
REVIEWER: L254 “These technologies are able to automatically study” the technologies don t study. You study, the technologies monitor, record, register, detect, survey..
AUTHORS: Completely agree. We have modified the manuscript accordingly.
REVIEWER: L259 Rephrase, grammar is wrong: “The obtained results shown a negative correlation between movement and temperature, however it may not always be so”
AUTHORS: We have modified the manuscript accordingly.
REVIEWER: L280 these results align with literature…
AUTHORS: Done.
Reviewer 2 Report
This manuscript involves a very interesting collaborative approach between a veterinary school and a computer science team. The approach used and described in this manuscript uses new technology (computer vision & image processing algorithm) to assess animal behavior or more precisely quantify the daily motion behavior and link that behavior with a disease. Although I found this MS very interesting and novel, there are several major issues that need to be addressed.
Major issues:
Based on the provided information in the statistical analysis section, the authors have used daily temperature reading & daily motion as independent measures (instead of repeated measures). If this is true, then this is a typical temporal pseudoreplication which occurs when an experimental unit is sampled repeatedly through time and the samples treated as if they represented independent experimental units. The authors need to clearly state what is the experimental unit and what data were gathered and used in the statistical analysis. Statistical analysis of such data without considering the data dependencies will only lead to erroneous results.
Methodological issue: there should be a clear definition of the objectives of this MS from the beginning, meaning the authors should state the objectives at the end of the introduction section. This will make the MS clear for the readers so they can follow the rest of the MS. As it is now there are several goals and aims of this MS that the authors wrote at different places in the MS e.g. lines 52-54, lines 84-85, lines 95-96.
Use of animals: It is not clear how many animals were involved in each experiment. The authors need to state clear how many animals were involved in this study so the readers can follow the methodology and results from this study.
The authors stated that the main goal was to identify if low motion is correlated with high temperature caused by fever (lines 84-85). Now this makes it unclear if this experiment is developed specifically to detect ASF or just quantify behavior related to any kind of fever. That said, then what is the justification to infect pigs with ASF for this kind of not specific disease experiment. Authors need to provide justification.
Lines 111-112: Please provide explanation about what criteria were used to discard some but not other pixels. I assume this will lead to errors when quantifying the movement variable. For this kind of work maybe a good idea is to validate the algorithm vs a gold standard which in this case will be a video recording. In that way there will be more certainty of whether the behavior measured by the algorithm correspondents with the observed behavior.
Section 2.3.3 Great explanation of the motion computations
Lines 192-193: both temperature and motion reduction are continuous variables so its unclear why authors performed Spearman’s correlation. Please explain.
Section 3.1. I don’t have expertise to comment on this part.
Lines 249-251: This is very strong conclusion, consider re-wording. I consider this as a pilot study that tested and improved the algorithm and showed how successfully it can quantify the motion of sick animals.
Author Response
Dear reviewer,
We thank you for your deep revision. We really think that all your considerations have been very beneficial to the improvement of the manuscript. According to your comments, we have modified the content taking into account other two reviewers’ considerations. Finally, we answer your inquiries below.
REVIEWER: Based on the provided information in the statistical analysis section, the authors have used daily temperature reading & daily motion as independent measures (instead of repeated measures). If this is true, then this is a typical temporal pseudoreplication which occurs when an experimental unit is sampled repeatedly through time and the samples treated as if they represented independent experimental units. The authors need to clearly state what is the experimental unit and what data were gathered and used in the statistical analysis. Statistical analysis of such data without considering the data dependencies will only lead to erroneous results.
AUTHORS: We agree that samples taken in the experiment are not independent, as they are finally taken from the same animal repeatedly. We thank the reviewer for noticing it.
First proposals involve mixed-models with fixed/random factors. However, far from exploring a real model, the correlation coefficient for repeated measures, proposed by Jonathan Z. Bakdash and Laura R. Marusich (“Repeated Measures Correlation”, 2017), which assesses the common intra-individual variance in data, should be enough for describing properly the correlation between temperature and motion. For further works, with more samples, a proper mixed-model will be carried out.
We have modified the manuscript accordingly.
REVIEWER: Methodological issue: there should be a clear definition of the objectives of this MS from the beginning, meaning the authors should state the objectives at the end of the introduction section. This will make the MS clear for the readers so they can follow the rest of the MS. As it is now there are several goals and aims of this MS that the authors wrote at different places in the MS e.g. lines 52-54, lines 84-85, lines 95-96.
AUTHORS: Accordingly, we have included a last paragraph in the Introduction section explaining the objectives of the MS.
REVIEWER: Use of animals: It is not clear how many animals were involved in each experiment. The authors need to state clearly how many animals were involved in this study so the readers can follow the methodology and results from this study.
AUTHORS: In “The experiments” subsections, we describe the characteristics of the experiments. Concretely, we describe the number of animals involved in each experiment: “The first experiment [...] two wild boar …” and “The second experiment [...] six wild boar…”. However, in order to ease the reading, we have also specified the number of animals involved in each experiment in key captions: Fig 6 and Table 1.
REVIEWER: The authors stated that the main goal was to identify if low motion is correlated with high temperature caused by fever (lines 84-85). Now this makes it unclear if this experiment is developed specifically to detect ASF or just quantify behavior related to any kind of fever. That said, then what is the justification to infect pigs with ASF for this kind of not specific disease experiment. Authors need to provide justification.
AUTHORS: Now, we have specified in the last paragraph of the Introduction that this work focuses on the increase of temperature caused by ASF virus: “The goal in this work is to show that animal motion behaviours change under high body temperatures caused by ASF virus.”. The whole MS is focused on ASF. However, in the Discussion, we have clarified the potential use in other diseases that cause fever: “The obtained results showed that computer vision systems can be used potentially to real-time febrile-disease surveillance.”.
REVIEWER: Lines 111-112: Please provide explanation about what criteria were used to discard some but not other pixels. I assume this will lead to errors when quantifying the movement variable. For this kind of work maybe a good idea is to validate the algorithm vs a gold standard which in this case will be a video recording. In that way there will be more certainty of whether the behavior measured by the algorithm corresponds with the observed behavior.
AUTHORS: Concretely, the criteria chosen might lead first to errors in the connected component creation, which may lead hereafter to animal recognition and motion computation. But not creating correctly the connected component of the collar, doesn’t disturb the CNN to recognize the collar properly, due to the CNN has been trained with a high enough quantity of images, even with errors in the connected components. In addition, in case that the animal cannot be recognized in one frame, generally, KF allows us to continue tracking the animal in the next frames till proper recognition. So, this step doesn’t interfere with the collar recognition, and consequently, with the motion computation.
In response to the question, the criteria in the morphological operations and in the blob analysis here applied are not pretty much remarkable. We applied standards in labeling connected components:
https://en.wikipedia.org/wiki/Connected-component_labeling
However, if the reviewer still considers a short explanation of this part in the text (M&M or Discussion). We can do it.
REVIEWER: Lines 192-193: both temperature and motion reduction are continuous variables so it's unclear why authors performed Spearman’s correlation. Please explain.
AUTHORS: This part has been reconducted towards a repeated measures correlation coefficient.
REVIEWER: Lines 249-251: This is a very strong conclusion, consider re-wording. I consider this as a pilot study that tested and improved the algorithm and showed how successfully it can quantify the motion of sick animals.
AUTHORS: Accordingly, we have softened the last two sentences. As suggested by the reviewer, this is a first work that should be continued to confirm any hypothesis.
Reviewer 3 Report
Dear authors
Congratulations for this extrordinary well prepared, organised and written MS! It was a pleasure to read it!
The analyses are well done. The idea is worthwile publication, as the idea should be developed further.
However, i have some concerns on the practability of the motion detection:It will be not possible to use it in the wild to detect ASF in wild boar populations. It can only be conducted on farms (housed pigs, free ranging pigs (e.g. Iberico), fenced hunting estates or wild boar farms). Nevertheless, this will be too late, as we need to detect ASF in the free ranging wild boar populations.
Thus you should stress these problems of adaptability and how the system could help detecting ASF in an early state MUCH more! Discuss the fact, that the system can only be conducted in enclosures much more. The detection of ASF in free ranging wild boar populations all over Europe would be nice to have, but seems not be possible even with motion detection. Needs to be discussed and should be mentioned in title and introduction (and abstract)
Otherwise the MS will just be an extraordnary well written MS on what an AI is able to do. And this should be published in a journal on computing...
specific remarks:
L34f: [...], Belarus (2013, in 2014, Estonia, Lithuania, Latvia, and Poland (all 2014) reported infection, [...]
L40f: And very recently Germany in September 2020!!! ASF in wild boar in the very east of Germany destroyed pork export from pig farms in the very west! Thus, even an eraly detection system in farms would not help much!
Fig 1: better replace 2018 by a map of 2018-2020?!
L98: typical spanish: repetition ;-) please delete this sentence
L99: ... is to recognise each blue or orange colored collar ...
Fig 2 and 3: why is there such a big space between figure and caption? ensure you hit all requirements and do not deliver large margins.
Fig. 6: Please decribe in Caption:
top: experiment E1, both animals are infected with ASFV Arm07
bottom: experiment E2, orange specimen (N=2) were infected with ASFV Arm07, blue specimen (N=4) were vaccinated with Lv17/wb/Rie1
Tab. 1: please add at end of caption "both infected with ASFV Arm07"
L250f: of course, this is right. However, how to implement in free ranging populations or at least on pig/wb farms?
L250: showed (the only error in English I found!)
Author Response
Dear reviewer,
We thank you for your deep revision. We really think that all your considerations have been very beneficial to the improvement of the manuscript. According to your comments, we have modified the content taking into account other two reviewers’ considerations. Finally, we answer your inquiries below.
REVIEWER: However, I have some concerns on the practicality of the motion detection:It will be not possible to use it in the wild to detect ASF in wild boar populations. It can only be conducted on farms (housed pigs, free ranging pigs (e.g. Iberico), fenced hunting estates or wild boar farms). Nevertheless, this will be too late, as we need to detect ASF in the free ranging wild boar populations.
AUTHORS: We totally agree. This tool is within the capacity for early detection, but mainly in pig farms that may have video surveillance. Likewise, this study shows some usefulness for surveillance in wild boar farms, which present a high risk of infection due to high mobility rates, translocations or releases. Accordingly, we have included a few lines to highlight the usefulness indoors but the required further work to fit this technology outdoors.
As indicated in the discussion, this study tries to present this tool for early detection of the disease regardless of the species studied. In this case we choose the wild boar and ASF as a model, but it is totally extrapolated to the domestic pig, as suggested in previous studies, without individual identification (Fernández-Carrión et al., 2017; Martínez-Avilés et al., 2017)
- Fernández-Carrión, E., Martínez-Avilés, M., Ivorra, B., Martínez-López, B., Ramos, Á. M., & Sánchez-Vizcaíno, J. M. (2017). Motion-based video monitoring for early detection of livestock diseases: The case of African swine fever. PloS one, 12(9), e0183793.
- Martínez‐Avilés, M., Fernández‐Carrión, E., López García‐Baones, J. M., & Sánchez‐Vizcaíno, J. M. (2017). Early detection of infection in pigs through an online monitoring system. Transboundary and emerging diseases, 64(2), 364-373.
REVIEWER: Thus you should stress these problems of adaptability and how the system could help detecting ASF in an early state MUCH more! Discuss the fact that the system can only be conducted in enclosures much more. The detection of ASF in free ranging wild boar populations all over Europe would be nice to have, but seems not to be possible even with motion detection. Needs to be discussed and should be mentioned in title and introduction (and abstract). Otherwise the MS will just be an extraordinary well written MS on what an AI is able to do. And this should be published in a journal on computing…
AUTHORS: The abstract, introduction and discussion have been modified accordingly.
REVIEWER: L34f: [...], Belarus (2013, in 2014, Estonia, Lithuania, Latvia, and Poland (all 2014) reported infection, [...]
AUTHORS: Modified.
REVIEWER: L40f: And very recently Germany in September 2020!!! ASF in wild boar in the very east of Germany destroyed pork export from pig farms in the very west! Thus, even an early detection system in farms would not help much!
AUTHORS: This information updates the manuscript. Thank you.
REVIEWER: Fig 1: better replace 2018 by a map of 2018-2020?!
AUTHORS: Done.
REVIEWER: L98: typical spanish: repetition ;-) please delete this sentence
AUTHORS: Done.
REVIEWER: L99: ... is to recognise each blue or orange colored collar ...
AUTHORS: Done.
REVIEWER: Fig 2 and 3: why is there such a big space between figure and caption? ensure you hit all requirements and do not deliver large margins.
AUTHORS: The margins depend on the template we used. I think the journal will suit better in the final step.
REVIEWER: Fig. 6: Please describe in Caption:
top: experiment E1, both animals are infected with ASFV Arm07
bottom: experiment E2, orange specimen (N=2) were infected with ASFV Arm07, blue specimen (N=4) were vaccinated with Lv17/wb/Rie1
AUTHORS: Done.
REVIEWER: Tab. 1: please add at end of caption "both infected with ASFV Arm07"
AUTHORS: Done.
REVIEWER: L250f: of course, this is right. However, how to implement it in free ranging populations or at least on pig/wb farms?
AUTHORS: The discussion section has been modified by explaining the implementation issue.
REVIEWER: L250: showed (the only error in English I found!)
AUTHORS: Done.
Round 2
Reviewer 2 Report
This MS is exploratory in nature and developed to test a new promising technology that uses computer vision and image processing to quantify sickness behavior in pigs.
The authors found a good way to overcome the dependency of the data for temperature and movement that was mentioned in Exp 1. However, both experiments have very few replicates, two animals in Exp1 and two Versus four animals in Exp2. Technically, it is possible to run some statistical analysis, however, the results from these analyses will be unreliable. Therefore, my suggestions to the authors is to drop all statistical testes mentioned in the MS and use descriptive statistics to report the findings they have.
Lines 59-62: Could a clarification be inputted somewhere in the introductions section
Line 197: “…some specific days were selected…” The readers will benefit if information is provided about how the authors selected the specific days for the analysis, including justification why some days were preferred over the other.
Lines 201-222: Please clarify and provide justification for the readers why non-parametric tests was used.
Lines 248-255: Based on this discussion point, it is unclear if this research was completed to test changes in behavior specific to ASF, or for other animals experiencing fever?
Author Response
Dear Reviewer,
we appreciate again your efforts on the review. We have modify slightly some parts of the MS, but we have some further questions previous to the final versions that we'd like to ask you.
Thank you.
